# Biological Effects of Animal Venoms on the Human Immune System

**DOI:** 10.3390/toxins14050344

**Published:** 2022-05-16

**Authors:** Zharick Avalo, María Claudia Barrera, Manuela Agudelo-Delgado, Gabriel J. Tobón, Carlos A. Cañas

**Affiliations:** 1Escuela de Bacteriología, Facultad de Salud, Universidad del Valle, Cali 76001, Colombia; zharick.avalo@correounivalle.edu.co (Z.A.); manuela.agudelo@correounivalle.edu.co (M.A.-D.); 2Department de Microbiología, Universidad del Valle, Cali 76001, Colombia; maria.claudia.barrera@correounivalle.edu.co; 3Department of Medical Microbiology, Southern Illinois University School of Medicine, Springfield, IL 62901, USA; gtobon56@siumed.edu; 4Unidad de Reumatología, Fundación Valle del Lili, Cali 760032, Colombia; 5Centro de Investigación en Reumatología, Autoinmunidad y Medicina Traslacional (CIRAT), Universidad Icesi, Cali 760031, Colombia

**Keywords:** envenomation, immunomodulatory, immunosuppression, innate immune system, adaptive immune system

## Abstract

Venoms are products of specialized glands and serve many living organisms to immobilize and kill prey, start digestive processes and act as a defense mechanism. Venoms affect different cells, cellular structures and tissues, such as skin, nervous, hematological, digestive, excretory and immune systems, as well as the heart, among other structures. Components of both the innate and adaptive immune systems can be stimulated or suppressed. Studying the effects on the cells and molecules produced by the immune system has been useful in many biomedical fields. The effects of venoms can be the basis for research and development of therapeutic protocols useful in the modulation of the immunological system, including different autoimmune diseases. This review focuses on the understanding of biological effects of diverse venom on the human immune system and how some of their components can be useful for the study and development of immunomodulatory drugs.

## 1. Introduction

Venoms, produced by animals, plants or microorganisms, are evolutionary products that serve to immobilize and kill prey, start digestive processes and act as a defense mechanism to prevent predation [1]. Their effects have been recognized and studied for many years as possible therapeutic strategies [2]. Venomous animals are important reservoirs of toxins, and they can be classified into two groups: those that produce their own toxins through gene expression and those that accumulate toxins and metabolites from the environment [3].

Venomous animals coevolved with humans are found worldwide and can be found on soil (scorpions, spiders, snakes, toads, salamanders, frogs, centipedes, ants, etc.), in the water (jellyfish, conical snails, sea anemones, corals, rays, etc.) and in the air (bees, wasps, hornets) [4]. These animals have specialized glands or cells that are attached to injection structures such as fangs, stingers and harpoons, through which they deposit venom after bite or contact [4].

Venom characterization is the scientific basis for research aimed at discovering beneficial uses of venom in biomedical science. Paracelsus (1493–1541) expressed the classic toxicology maxim “The dose makes the poison”, which is a principle that suggested that the beneficial effects of biological changes can be determined by the amount of the substance used [5]. Charles Lucien Bonaparte was the first to establish the proteinaceous nature of snake venom in 1843 [6]. In the late eighteenth century, Felice Fontana studied the effect of snake venom on coagulation [7], and in the nineteenth century, with the production of anti-venom sera [8], the relationship between envenomation and the immune system was considered. Many components of animal venoms are known to trigger the production of proinflammatory mediators, modulating secondary effectors of cell signaling, such as protein kinases, and altering extracellular matrix components and receptors [9,10]. The inhibition of the immune response has also been described [11]. Although these venoms can be harmful or even lethal when activating the immune system, their dose-dependent effects or the effects of inhibitors can serve as therapeutic tools in a variety of clinical conditions [12,13].

This review focuses on understanding the activators and modulatory effects of venom components on the human immune system and how some of these venoms can be potential therapeutic tools.

## 2. Effects on Innate Immunity

Innate immunity is the first line of defense against microorganisms and external stimuli; it is nonspecific and is composed of physical barriers, soluble antimicrobial peptides and molecules, effector cells and phagocytes [14]. Soluble molecules are useful in opsonization, inflammation and pathogen clearance, while the cellular components are responsible for both elimination and the stimulation of adaptive immunity through antigenic presentation [15,16,17]. This type of immunity can result in a fast and effective response without prior recognition of the stimulus [18].

### 2.1. Effects on Phagocytes

Studies have attempted to elucidate the innate immune pathways that are modulated by exposure to venom components, which alter phagocytes mainly by interacting with membrane receptors or intracellular signaling pathways. Diverse venom components interact with receptors on inflammatory cells, activating them and triggering the establishment of a proinflammatory state, which is accompanied by mild anti-inflammatory compensation that attempts to reverse associated local and systemic effects [12].

The initial stimulation of phagocytes is mediated by the recognition of molecules present in the venom by Toll-like receptors (TLRs), especially TLR2 and TLR4. Studies have shown that TLR2 and TLR4 recognize the Ts1 toxin from the venom of the scorpion *T. serrulatus* [19]. This β-toxin binds to TLR2 and TLR4, inducing the production of cytokines and lipid mediators in a MyD88-dependent manner, leading to the activation of proinflammatory signaling, such as the NF-κB, AP-1 or MAPK pathways (ERK1/2 and MAPK p38) [19]. The greatest toxic effect of Ts1 occurs at the level of sodium channels [20,21,22,23]. MT-III is a group IIA secreted phospholipase A2 (sPLA2) present in *Bothrops atrox* and *Bothrops asper* snake venom and is also recognized by TLR2, leading to the production of eicosanoids such as prostaglandin E2 (PGE2) and leukotriene B4 (LTB4), which are potent neutrophil chemoattractants that favor the resolution of the acute response at the injury site [24,25]. Other studies have shown the role of TLR2 in stimulating polymorphonuclear migration and the production of interleukin (IL)-1β and inhibiting the production of IL-6 and mononuclear migration to the injury site in murine models with an intraperitoneal injection of venom from the snake *B. atrox* [25]. This finding is important because it demonstrates the effect of venoms on proinflammatory signaling pathways through interactions with receptors and their potential use as therapeutic agents.

Crotoxin (CTX) is a major component of the venom of snakes in the genus *Crotalus* and stimulates receptors such as formylated peptide receptors and muscarinic receptors [26,27]. *C. durissus terrificus* CTX has been shown to promote anti-inflammatory effects by interfering with leukocyte migration in murine models [26]. Studies have shown that this anti-inflammatory effect of CTX is dependent on its interaction with formylated peptide receptors since the use of antagonists of these receptors, such as Boc-2, inhibited this effect on dendritic cells and macrophages [27,28].

The CTX of *C. durissus* venom is composed of two subunits: CA, also known as crotapotin, and CB, which is a weakly toxic phospholipase A2 with high enzymatic activity. The CB fraction of CTX can decrease the expression of MHC type II molecules, which are important for antigen presentation to T-lymphocytes, as well as costimulatory molecules such as CD40, CD80 and CD86 [28]. Additionally, the toxin inhibits the production of IL-6, TNF-α and IL-12 by interfering in the phosphorylation of NF-κBp65, ERK1/2 and MAPK p38 [28]. On the other hand, the CB fraction stimulates the production of IL-10, TGF-β, PGE2 and LXA4, thus exerting an anti-inflammatory effect [28,29]. Envenomation by *C. durissus*, compared to that of other vipers, generates a smaller inflammatory reaction and less pain, which has drawn the attention of researchers looking for anti-inflammatory and analgesic elements of this toxin [12].

In macrophages, the effect of CTX on phagocytic propagation and activity has been characterized, and toxin-mediated alterations in cytoskeletal proteins modulate phagocytosis independent of the receptor involved. The same toxin can stimulate the production of nitric oxide (NO) by activating the enzyme inducible nitric oxide synthase (iNOS) and the NO-GMP pathway [30]. Glucose and glutamine metabolism is also altered by the hyperactivity of hexokinase, glucose 6-phosphate dehydrogenase and glutaminase [30,31]. Enzymatic hyperactivity in macrophages is associated with increased levels of ATP and metabolites from biosynthetic pathways and the stimulation of inflammatory and immune responses, mainly through the production of NADPH [30]. NADPH serves as a substrate for NADPH oxidase, removing electrons from NADPH to reduce O_2_ to O_2-_, which is rapidly converted to H_2_O_2_ [32]. The effect on macrophages was shown to be prolonged, as it could be detected in circulating macrophages up to 7 days after treatment with CTX [30].

On the other hand, the crude venom of *C. durissus terrificus* stimulated the production of TNF, IL-6 and interferon (IFN)-γ, which peaked between 24–72 h after intraperitoneal injection in mice [33].

The effects of the fractions and crude venom of some scorpions on macrophages have also been studied. The FII and FIII fractions of Ts1 and Ts6 of *T. serrulatus* venom dose-dependently stimulated the production of NO and H_2_O_2_ in macrophages. These toxins also modulate the inflammatory response by increasing levels of TNF-α, IL-6, IL-1α, IL-1β, IL-8 and GM-CSF, which trigger systemic inflammation and increase the risk of complications [34,35,36,37]. Additionally, Ts2 toxin exerts anti-inflammatory effects through the production of IL-10 [38].

Macrophage differentiation into subpopulations is also modulated by toxins from other animals, such as *Androctonus australis hector*, a species of scorpion widely distributed in Africa and Asia. Its venom (AahV) has two toxins called AahI and AahII that have regulatory effects on macrophage polarization to the M1 phenotype by stimulating the expression of inflammatory genes such as Il1b, Il23 and nos2 and dysregulating the expression of genes associated with the M2 phenotype, such as Arg1 and Il10 [39].

Venoms can affect the migration of macrophages; pretreatment with *Apis mellifera* venom has been studied in models of experimental autoimmune encephalitis in which inhibition of the mRNA expression of macrophage chemoattractants such as RANTES, MCP-1 and MIP-1α has been demonstrated [40].

In neutrophils, *Loxosceles* spider venom causes its indirect activation by an endothelial-mediated mechanism [41]. *Loxosceles* venom contains phospholipases D, which hydrolyze sphingomyelin and subsequently liberated ceramides act as intermediaries that regulate TNF-α and recruit neutrophils [42]. Inhibition of actin polymerization, tyrosine phosphorylation, and RhoA and Rac1 protein activity has been demonstrated, leading to the inhibition of phagocytosis for up to 14 days after treatment with CTX [43]. Additionally, CTX, the TzII and TzIII toxins of the scorpion *T. zulianus* and the venom of the snake *Calloselasma rhodostoma* have positive effects on the production of reactive oxygen species (ROS) and the release of myeloperoxidase, thus contributing to the development of neutrophil effector functions [44].

The venom of aquatic animals has made it possible to modulate the effects of neutrophils on innate immunity; venom isolated from the freshwater ray *Potamotrygon* cf. *henlei* induced neutrophilia that depended on TLR/TRIF signaling that was stimulated by the release of IL-33 derived from cells at the site of injury [45]. Likewise, the venom of the saltwater ray *Hypanus americanus* has been shown to induce strong swelling and leukocyte infiltration in the legs in murine models, which indicates that the venoms of marine animals probably contain molecules or toxins that promote this inflammatory state [46]. The venom of aquatic animals differs in composition from the venom of land animals and is characterized by the presence of phosphatidylcholine-2-acylhydrolase, metalloproteinases and hyaluronidases; these molecules degrade the extracellular matrix and induce the production of IL-33 in epithelial and endothelial cells [45]. The binding of IL-33 to the ST2 receptor stimulates MyD88-dependent MAPK signaling, resulting in the phosphorylation of ERK 1/2, MAPK p38, JNK and NF-κB. This amplification of the innate immune response and the generation of a microenvironment that leads to the accumulation of neutrophils at the injury site promotes tissue damage and necrosis due to the release of ROS, NO, myeloperoxidases such as MMP-9, serine proteases and cathepsin G, among others [45,47]. IL-33 production affects aryl hydrocarbon receptors present on mast cells, which also respond to environmental toxins and endogenous components, and triggers the production of IL-17 and ROS by these cells [48]. Increased IL-17 production is associated with the overexpression of CXCL5, which is a potent neutrophil chemoattractant [49]. Likewise, the venom of *T serrulatus* (TsV), the CTX of *C. durissus terrificus*, the batroxase and BatroxPLA2 of *B. atrox,* and the piratoxin-I, bothropstoxin-I and bothropstoxin-II of *Naja mocambica* have also been associated with the modulation of cell migration [50].

Regarding the action of venoms on bone marrow and hematopoietic structures, studies are contradictory. In the case of *Tityus serrulatus* venom, there are conflicting findings in mice studies. Both stimulation [51] and inhibition [52] have been reported. Crotoxin, rattlesnake toxin, down-modulates functions of bone marrow neutrophils and impairs the Syk-GTPase pathway [53]. Extensive studies in this regard are lacking.

### 2.2. Effects on Mast Cells

Mast cells are also activated by toxins from animal venoms, which involves excessive activation of the inflammasome and increased expression of caspase 1, which demonstrates a protective effect against the venom [54]. Additionally, mast cells respond to toxins such as LmTx-I from the snake *Lachesis muta,* which trigger the production of histamine and lipid mediators, favoring inflammation [55]. Mast cell activation and degranulation are also stimulated by CTX of *C. durissus terrificus*, *C. durissus cascavella* and *C. durissus collilineatus* [56]. There is strong evidence of an association between severe anaphylaxis, especially hymenoptera venom-induced anaphylaxis, and mast cell disorders. It has been thought that intrinsic abnormalities in mast cells, including the presence of the activating KIT D816V mutation in mastocytosis or of genetic trait, hereditary alpha-tryptasemia, may influence susceptibility to severe anaphylaxis. The understanding of these mechanisms in susceptible individuals can shed light on the expansion of knowledge on anaphylaxis and its treatment [57].

### 2.3. Effects on Complement

Another function of *B. atrox* is to induce the generation of the complement fractions C3a and C5a, which are potent anaphylatoxins that promote mast cell degranulation and neutrophil chemotaxis and activation [53,58].

The complement system is activated in the presence of toxins from snakes, such as cobras in the *Naja* genus and the G2 fraction of flavoxobin from *Trimeresurus flavoviridis;* the former is a source of a protein similar to the C3 protein of complement, which is called cobra venom factor (CVF). This factor gives rise to an enzyme called CVFBb, which is highly stable and has C3/C5 convertase activity, and the second promotes C3a release and the assembly of the membrane attack complex [56,59,60]. The modulation of complement activation pathways has been studied, and the toxins BjussuSP-I from *B. jararacussu* and BpirSP27 and BpirSP41 from *Bothrops pirajai* are inhibitors of the classic and lectin pathways and slightly modulate the alternative complement pathway [61].

The study of complement depletion by CVF has allowed elucidation of the behavior of this system in the development of the innate immune response and in the pathogenesis of some inflammatory diseases [12].

A comprehensive review of the effects of the various venoms on the complement system has been published [62].

### 2.4. Effects on Cytokines

Inflammation is characterized by redness, swelling, warmth, pain and loss of function in the affected tissue, which are consequences of immune cell responses and the vascular and inflammatory events associated with infection or injury [14,63]. Changes at the circulatory level are related to changes in vascular permeability, leukocyte recruitment and infiltration and the release of inflammatory mediators and cytokines [64].

The immune system responds to tissue damage through the initiation of a chemical signaling cascade to repair the affected tissues. This type of chemical signal is necessary for leukocyte chemotaxis to the site of injury, where activated leukocytes are responsible for resolving the inflammatory response through the production of cytokines, chemokines and lipid mediators [14].

Some cytokines are modulated by toxins. The venom of *Centruroides noxius,* which is an important scorpion species associated with multiple attacks in Mexico, exerted an important in vivo effect on the secretion of TNF-α, IL-6 and IFN-γ, which was maintained for up to 21 days [65]. After 21 days, there is an increase of IL-10, suggesting the modulation from an anti-inflammatory profile of other components of the venom [65].

The venom of *A. australis hector* and the toxins Bl-PLA2 and Bbill-TX from snakes *B. leucurus and B. billilineata* have been shown to increase the levels of proinflammatory cytokines such as IL-4, IL-6, IL-12, TNF-α and IL-1β, which are associated with hemolytic activity and leukocytosis due to neutrophilia and eosinophilia [66,67,68,69]. The cytokines IL-1, IL-6 and IL-12 have multiple functions and act synergistically to establish acute inflammation in tissues and modulate the function and differentiation of T- and B-lymphocytes [67].

Modulation of the anti-inflammatory profile has been evaluated in models of envenomation with the venom of *Crotalus durissus collilineatus*, *Daboia russelii*, *C. durissus terrificus* and species of *Bothrops* spp., and the results showed an increase in the production of IL-10 [70,71,72].

The gene expression of cytokines has been studied in response to venoms from other animals, such as that of the viper *Vipera ammodytes ammodytes*, and it was shown that the venom of this snake stimulates the expression of proinflammatory genes such as *Il1a*, *Il1b*, *Ifna2 and Ifnb1* [73]. The same venom is capable of downregulating the production of IL-12 and IL-18, which are potent stimulators of IFN-γ release [73]. IL-12- and IL-18-induced production of IFN-γ is important for inducing the cytotoxic activity of innate immune cells, as well as for the development and maintenance of the Th1 response [74]. Similarly, the venoms of *V. ammodytes ammodytes*, *B. billilineata* and *Calloselasma rhodostoma* stimulate the gene expression of IL-8 by neutrophils, which is a powerful chemoattractant of polymorphonuclear cells, CD4+ and CD8+ T-lymphocytes and NK cells [43,75].

The urine of mice that were intraperitoneally injected with *C. durissus terrificus* venom showed increased concentrations of TNF-α, IL-10, IL-5, IL-6 and IFN-γ 15 min to 4 h after inoculation, and these levels decreased in urine after 4 h [76,77]. These findings were associated with kidney damage characterized by proteinuria and elevated serum creatinine levels [77]. The renal alterations are related to the accumulation of proinflammatory cytokines, which induce the rupture and desquamation of the tubular epithelium, favoring the development of proteinuria and the loss of renal function [77,78].

The effects of toxins isolated from the venom of bees and snakes have on the progression and development of symptoms in inflammatory and autoimmune disorders such as rheumatoid arthritis (RA) [79], acute intestinal inflammation [80], systemic lupus erythematosus (SLE) [81] and systemic inflammatory syndrome [82] have been evaluated. Thus, many modulatory effects have been studied, including the modulatory effects of cytokines.

Cobratoxin, cardiotoxin and neurotoxin derived from *N. naja atra* venom exert anti-inflammatory responses by reducing the levels of TNF-α and IL-1β and the enzymatic activities of myeloperoxidase (MPO) and iNOS [83,84]. This anti-inflammatory effect has also been demonstrated by the poisons of *Apis mellifera* and the cobra *Naja naja atra*, which have been evaluated as possible treatments for rheumatoid arthritis since the administration of different concentrations of these agents in murine models reduced edema in the legs, as well as the arthritis index and inflammatory pain [85]. In addition, bee venom contains PLA2, melittin and hyaluronidases, among other factors. Bee venom is useful in decreasing serum levels of rheumatoid factor, PCR, anti-streptolysin O and proinflammatory cytokines such as TNF-α, IFN-γ, IL-6 and IL-1β [79,86]. The excretion of hydroxyproline and glucosamine in urine, as well as serum levels of acid phosphatase and alkaline phosphatase, was also decreased in RA models treated with *Naja kouthia* venom, indicating the protective and anti-inflammatory effect on cartilage degeneration and destruction [87]. The action of diverse components of venoms on the extracellular matrix (both in the damage and in the reparative processes) is a fascinating field of research that can contribute to the knowledge of the pathogenesis and possible treatment of connective tissue diseases and cancer [88]. This field of knowledge is outside the scope of this work, but it must be taken into account.

CTX has also been shown to be effective in dysregulating acute intestinal inflammation in BALB/c mouse models of 2,4,6-trinitrobenzene sulfonic acid (TNBS)-induced colitis through toxin-mediated inhibition of IL-17A and IFN-γ secretion by innate lymphoid cells (ILC) 1 and ILC3 located in the intestinal mucosa [80]. The effect of the venom of the *Naja naja atra* cobra on systemic lupus erythematosus was evidenced by the decrease in lymphadenopathy, skin erythema, proteinuria and tissue lesions in MRL/lp mice. In addition, the production of TNF-α and IL-6 and the levels of anti-dsDNA antibodies decreased [81]. The decrease in this type of antibody is important due to its association with renal deterioration and the activation of dendritic cells and, thus, the development of the adaptive immune response [89].

### 2.5. Effects on Inflammasomes

The inflammasome is a multiprotein complex that participates in the inflammatory response against venoms through the production of the cytokines IL-1β and IL-18, tissue repair and pyroptosis [90,91]. The ability of *Apis mellifera* venom to activate the NLRP3 inflammasome has been demonstrated both in vivo and in vitro, and its effect is mainly attributed to melittin in the venom, which forms pores in the cells, generating a signal that is recognized by the immune system and activates the inflammasome [54].

### 2.6. Effects on Transcription Factors

It is important to consider that the inflammatory response is regulated by various transcription factors; in physiological conditions, NF-κB binds to the inhibitory protein IκB (IκB-α, IκB-β, IκB-ε, etc.) in the cytoplasm [92]. Under inflammatory conditions, some cytokines, such as TNF-α and IL-1β, promote the phosphorylation of kinases of IκB proteins, which are degraded and release NF-κBp65 to translocate to the nucleus and initiate the transcription of proinflammatory cytokine genes [92,93]. Cobratoxin from *Naja naja atra* decreased the levels of phosphorylated IKK-α and phosphorylated IκB-α, blocking the translocation of NF-κBp65 to the nucleus [94]. Cobratoxin from other snakes, inhibited the NF-κB pathway by binding with high affinity to the IKK proteins in the canonical pathway; these proteins are involved in the phosphorylation and degradation of IκB, blocking the release of NF-κB dimers and subsequent nuclear translocation [95]. In addition, the venom of other animals, such as the parasitoid wasp *Nasonia vitripennis*, suppresses the inhibitors IκBα and A20, inhibiting negative feedback and the activity of NF-κB [96]. In addition, venom stimulates the transcription of glucocorticoid-regulated genes such as GILZ and MKP1, which inhibit NF-κB by interacting with p65 and dephosphorylating and inactivating JNK and p38 [97,98,99,100]. The effect of the venom was also evidenced in the prolongation of JNK activation in the MAPK pathway in LPS-induced macrophages, which could lead to the development of a cytotoxic response in the cell when high concentrations of the venom were administered [96]. The inhibitory effect of other poisons, such as the *Apis mellifera* venom and the *N. naja atra* venom, on inflammatory intracellular signaling decrease the release of IL-17 by reducing the phosphorylation of p38 in the MAPK pathway, as well as reducing the activation and translocation of NF-κB to the nucleus [40,83].

## 3. Effects on Adaptive Immunity

The adaptive immune system is made up of cellular and humoral components. The cellular components include immune effector cells such as T-lymphocytes and B-lymphocytes. Furthermore, the humoral components include the different isotypes of immunoglobulins or antibodies produced by B-lymphocytes [14,101].

The key functions of adaptive immunity are the differentiation of self-antigens or molecules from those that are not self, the generation of targeted immune responses to eliminate or neutralize pathogens or infected cells, the development of immune memory and the recovery of homeostasis. The hallmark of adaptive immunity is the ability of the system to generate a memory of pathogens or harmful stimuli, which allows the body to mount a faster and more effective immune response during subsequent exposures to this antigen or stimulus [101].

The effects of venoms on the components of adaptive immunity have also been tested to assess their modulation of responses and consider possible therapeutic targets for immune disorders.

### 3.1. Effects on B-Lymphocytes

*C. durissus terrificus* venom and CTX decreased IgG1 and IgG2a levels in BALB/c mice immunized with human serum albumin (HSA) and ovalbumin (OVA), and this effect was independent of the adjuvant used for immunization [29,102]. This phenomenon is attributed to the inhibitory effect on activation signals such as IL-4, as well as effects on the CD40-CD40L interaction between Th2 lymphocytes and B-lymphocytes, which is necessary for the correct development of humoral immunity [103]. Suppression of IL-4 production is important, given its role in Th2 differentiation, as well as stimulation of MHC class II expression and mitogenic effects on B-lymphocytes through the JAK/STAT pathway by transduction of the transcription factor STAT3 or STAT6 [74].

However, the venom of other snakes, such as *Naja naja atra* stimulated the production of antibodies against ram erythrocytes (SRBCs) in immunotoxicity assays by stimulating the production of IL-4 [104,105]. *N. naja atra* venom also had a positive regulatory effect on germinal centers in animal immunosuppression models, restoring their architecture and promoting the production of IgM and IgG [105].

### 3.2. Effects on T-Lymphocytes

CTX also affects the depletion of lymphocytes in peripheral blood and plasma by stimulating lymphocyte expression of adhesion molecules, such as LFA-1 (integrin alpha L Betha 2) and VLA-4 (integrin Alpha4 Beta1), which bind to endothelial cells via their specific ligands ICAM-1 and VCAM-1, respectively; this effect is attributed to the overregulation of lipoxygenase-derived mediators [106]. Lymphocyte depletion in peripheral blood was also observed in response to cardiotoxin III of *Naja naja atra* venom, which decreased the numbers of CD4+ and CD8+ T-lymphocytes by stopping the cell cycle in the G0/G1 phase [86,105].

Genotoxicity is defined as the destructive effect on genetic material (DNA, RNA) by altering their integrity, to which the cell responds by genetic repair mechanisms or apoptosis [107]. Lymphocytes were exposed to the venom of *Apis mellifera* and showed high genotoxicity independent of dose or time in studies of kite trials [108]. This genotoxicity was accompanied by the induction of apoptosis by activating pro-apoptotic caspase 3 and deregulating anti-apoptotic molecules such as Bcl-2, ERK and Akt, which are involved in survival cell-mediated activation of NF-κB [108,109]. Toxins from various *Bothrops* species, mainly BthTX-I and BthTX-II from *B. jararacussu* and BatxLAAO from *B. atrox*, have also shown genotoxic effects on peripheral blood lymphocytes [110].

Effective immunotherapy with bee venom has been shown to improve the response of individuals with systemic reactions to bee stings, and the effect is largely attributed to the stimulation of regulatory T-lymphocyte (Treg) differentiation and IL-10 production, which generates an immunologically tolerant state in the individual [111,112,113]. Treg differentiation may or may not include the expression of the lymphocyte activation marker CD25; however, Foxp3 expression is induced by high-avid interactions of class II self-peptide/MHC complexes. Strong TCR signaling and suboptimal costimulation, as well as how high amounts of TGF-β and retinoic acid, determine the generation of this subpopulation among both thymocytes and peripheral naive CD4 T-lymphocytes [114]. After repeated exposure, bee venom promotes the differentiation of naive CD4+ T-lymphocytes and mature thymocytes into the IL-10-producing Treg subpopulation by increasing the expression of markers such as CD25+ and Foxp3+ [113,115]. In addition, the suppression of the venom-induced T cell response is due to the stimulation of histamine type II receptors on Th2 lymphocytes and to the increased IL-2 production and expression of CTLA-4 and PD-1 in CD4+ T-lymphocytes, which are important for inhibiting initial self-reactivity during the activation of T cells in lymph nodes and peripheral tissues, respectively [113,116]. Likewise, the effect of bee venom on CD4+ T-lymphocyte subpopulations has also been examined in models of atopic dermatitis, and the results showed a decrease in the production of Th2 and Th1 cytokines [117].

The immunoregulatory effects of bee venom may play roles in research aimed at immunomodulation of pathologies such as multiple sclerosis and amyotrophic lateral sclerosis; this effect is attributed to the decrease in microglial activation and infiltration of Th1 lymphocytes (CD4+ and IFN-γ) and Th17 cells (CD4+ and IL-17) and the favorable recruitment of Tregs [40,118,119].

Other toxins have been shown to stimulate the differentiation of lymphocyte subpopulations, such as those derived from the conotoxins of the marine snail *Californiconus californicus* called cal14.1b and cal14.2c [120]. These toxins interact with the α7 subunit of nACh receptors, which are abundantly expressed on Treg lymphocytes [120]. This interaction decreases the production of IL-17A and inhibits NF-κB signaling; therefore, this interaction has an impact on the production of cytokines such as IL-2, IFN-γ, IL-4 and TNF-α, inhibiting the Th1 and Th17 lymphocyte-dependent inflammatory responses that promote the development of cellular immunity and the risk of T cell-mediated autoimmune diseases [40,105,120].

This same inhibitory effect on the Th17 subpopulation has been demonstrated in studies with *N. naja atra* venom and *C. durissus terrificus* CTX, which promote the expression of Foxp3 on CD4+CD25+ and CD4+CD25− lymphocytes and suppress IL-17 production [80,105,120,121].

The K^+^ channels on lymphocytes are targets of toxins such as Ts1, Ts2 and Ts6 of the venom of *T. serrulatus* and the toxin MgTX of the venom of *Centruroides margaritatus*, which suppress the expansion of T-lymphocytes and IL-2 production [122]. Pharmacological blockade of KV1.3 channel inhibits Ca^2+^ signaling, T cell proliferation, and pro-inflammatory interleukins production in human CD4+ effector memory T cells [123].

Melittin, the major pain-producing substance of honeybees (*Apis mellifera*) is cytotoxic for T-lymphocytes and able to induce morphological changes in the cell membrane, granulation and lysis [124]. In addition, they show increased DNA damage, including oxidative DNA damage, as well as the increased formation of other markers of genomic instability [125]. This genotoxicity coincides with the increased formation of reactive oxygen species, reduction of Glutathione, increased lipid peroxidation as well as Phospholipase C activity, showing the induction of oxidative stress [126]. Melittin itself is also capable of modulating gene expression patterns of genes involved in DNA damage response, oxidative stress and apoptosis [127].

## 4. Effects on Binding to Extracellular Matrix

Multicellular organisms require the integration of cells with each other and with the extracellular matrix through proper adhesion. Cell adhesion molecules are grouped into four important families: cadherins, some members of the immunoglobulin superfamily, selectins and integrins. With the exception of mature erythrocytes, all cell types have one or more integrins expressed on their surface. In mammals, approximately 20 different integrins can be found [128]. Disintegrins in snake venom act by stimulating and inhibiting integrins, altering their natural adhesion to molecules of the immunoglobulin family or to extracellular matrix structures.

α1β1 (VLA-1, CD49a/CD29) integrin is localized at the T-lymphocytes, and their ligands are laminin and collagen, jointing to lysine-tryptophan-serine (KTS) and arginine-tryptophan-serine (RTS) motifs and inhibited by Obtustatin [129,130] and Viperistatin [131] from *Vipera lebetina obtuse.* α2β1 (VLA-2, Ia/IIa, CD49b/CD29) are located in both T- and B-lymphocytes and bind to laminin and collagen; they are inhibited by Rhodocetin [132] and Bilinexin [133] of *Calloselasma rhodostoma* and *Agkistrodon bilineatus*, respectively. α4β1 (VLA-4, CD49d/CD29) is located in both T- and B-lymphocytes and binds to fibronectin (VCAM-1 via methionine-leucine-aspartic acid-MLD motif) and is inhibited by EC5 and VLO5 toxins of *Echis carinatus* and *V. lebetina obtuse* [134]. α5β1 (VLA-5, CD49e/CD29) of T-lymphocytes binds to fibronectin (arginine-glycine-aspartic acid-RGD motif) and is inhibited by Trigramin [135] and Contortrostatin [136,137,138] of *Trimeresurus gramineus* and *A. contortrix contortrix* respectively.

## 5. Biomedical Applications of Venom in Autoimmunity-Special Attention to Snake Venoms

Venoms have been useful for the development of biomedical applications such as the study of the pathogenesis of various diseases, the design of diagnostic tests and the design of new drugs, including immunosuppressors. Through this review, some approximations of biotechnological applications of venoms have been given; some concepts are expanded below, given their importance.

Snake venom disintegrins have played an important role in the study of integrins, such as their location and function [139]. Their possible therapeutic applications derive from their physiological impact on human integrins. Integrins have been studied for their possible use as antineoplastic drugs, mainly due to their alleged inhibitory action of neoplastic cell adhesion to the extracellular matrix, a process that would prevent the development of metastasis [140,141,142,143,144]. Given this mechanism of action, it can be hypothesized that they could also be useful for the development of anti-inflammatory drugs by slowing cell infiltration into tissues.

Contortrostatin from *A. contortrix contortrix* has additionally been studied because of its antiangiogenic effect [145,146,147]; Salmosin [148] from *A. brevicaudus* because of its antiangiogenic and apoptosis-inducing effect; Bitistatin [149] from *Bitis arietans* and *Echistatin* [150] from *E. sochureki carinatus* have, in addition to an antiangiogenic effect, other effects that alter intracellular dynamics by inhibiting cellular function. These actions are also useful for the development of antineoplastic and immunosuppressive drugs. Rhodostomin from *C. rhodostoma* alleviates inflammatory response based on sepsis models through Integrin alphaVbeta3 blockade [151]. Trimucrin from *T. mucrosquamatus* suppresses LPS-induced activation of phagocytes primarily through blockade of NF-κB and MAPK activation and is postulated as anti-inflammatory [152].

Crotoxin, as previously discussed, is a β-neurotoxin that is the main toxic component of the *C. durissus terrificus* snake, capable of inducing neuromuscular paralysis, cardiorespiratory failure and potentiating the effect of FLA2 [153,154]. Ophidiotoxicosis by this venom, compared to that of other vipers, generates a lower inflammatory reaction and less pain, a fact that has drawn the attention of researchers to look for anti-inflammatory [155] and analgesic [156] elements in Crotoxin. The immunogenicity of Crotoxin is low, which has led researchers to consider the existence of an immunosuppressive effect [157,158].

The anti-inflammatory effect of Crotoxin is evidenced in experimental models of inflammation in animals where the production of anti-inflammatory cytokines (IL-10 and IL-14) [159] is induced, as well as the inhibition of phagocytosis by macrophages [160] and neutrophils [161] and the effects on cell migration, inhibiting interaction with the endothelium [162]. Crotapotin, the Crotoxin complex acid subunit, which lacks enzymatic or neurotoxic activity, and acts as a chaperone, inhibiting the effect of FLA2; it is in this sense that its role as an anti-inflammatory is considered. The inhibitory effect of Crotoxin on the components of antigen presentation is also considered [163].

The so-called CVF, isolated from *N. naja* [164], has a complement-activating effect and has been used in the study of the complement cascade. The Oxiagin extracted from *N. oxiana* venom inhibits the formation of C3 convertase by inhibiting the classical complement pathway. This finding is very important because of its potential for the development of inhibitory drugs for this action, which is key in the origin of various diseases, including autoimmune diseases.

Table 1 shows a summary of the relevant effects of some venoms on the immune system found in this study with reference to the species from which the venoms are obtained. Possible biotechnological applications are shown.

## Figures and Tables

**Table 1 toxins-14-00344-t001:** Animal species whose venoms have been studied for their effects on the immune system and the most relevant effects in this system. Possible applications are shown.

Animal (Venom From)	Description (Reference)	Possible Applications
*Androctonus australis hector*	Polarization from macrophages to the M1 subpopulation [39].	Study of macrophage dynamics
	Elevation of pro-inflammatory cytokines such as IL-4, IL-6, IL-12, TNF-α and IL-1β [66].	Study of forms of induction of cytokine release
*Apis mellifera*	Inhibition of phagocytosis [40]. Reduction of TNF-α, IL-1β levels, and the enzymatic activities of myeloperoxidase (MPO) and iNOS synthetase [85].	Drugs to stop action of phagocytes. Ex. Hemophagocytic síndrome Anticytokine therapies
	Promotes a differentiation of naive CD4+ T-lymphocytes and mature thymocytes in the IL-10-producing Treg subpopulation [115]. Decrease of the production of Th2 and Th1 profile cytokines [118]	Immunosuppressive therapy
*Bothrops asper*	Production of eicosanoids such as PGE2 and LTB4 [18].	Prostaglandin activation study
*Bothrops atrox*	Modulation in cell migration [25]. Increase IL-1β IL-1B, IL-10, IL-6, TNF- α and eicosanoids release [25,54].	Anti-inflammatory therapy
	Generation of C3a and C5a anaphylatoxins [57].	Study of complement system
	Genotoxic potential on peripheral blood lymphocytes [110].	Human genome damage study
*Bothrops bilineata*	Elevation of pro-inflammatory cytokines such as IL-4, IL-6, IL-12, TNF-α and IL-1β [69].	Study of forms of induction of cytokine release
	Stimulate the gene expression of IL-8 by neutrophils [75].	Neutrophil chemotaxis study
*Bothrops jararacussu*	Alteration of the classical, alternative and lectin pathways of the complement system [62].	Study of complement system
	Genotoxic potential on peripheral blood lymphocytes [110].	Human genome damage study
*Bothrops leucurus*	Elevation of pro-inflammatory cytokines such as IL-4, IL-6, IL- 12, TNF-α and IL-1β [68].	Study of forms of induction of cytokine release
*Bothrops pirajai*	Alteration of the classical, alternative and lectin pathways of the complement system [61].	Study of complement system
*Californiconus californicus*	It decreases IL-17A production and shuts down NF-kB signaling [120].	Immunosuppressive therapy
*Calloselasma* *rhodostoma*	Reactive oxygen species and myeloperoxidase release [43]. Stimulate the gene expression of IL-8 by neutrophils [44].	Neutrophil activation and neutrophil chemotaxis studies
*Centruroides margaritatus*	Suppression of T-lymphocytes expansion, protein synthesis and IL-2 production [122].	Immunosuppressive therapy
*Centruroides noxius*	Secretion of TNF, IL-6 and IFN-γ. Subsequent elevation of IL -10 [35].	Study of cytokine dynamics
*Crotalus durissus*	Stimulation of formylated peptide receptor involved in chemotaxis [26].	Study of chemotaxis
*Crotalus durissus cascavella*	Mast cell activation and degranulation [57].	Study of mast cell dynamics
*Crotalus durissus collilineatus*	Mast cell activation and degranulation [57].	Study of mast cell dynamics
	Anti-inflammatory response by IL-10 elevation [70].	Immunosuppressive therapy
*Crotalus durissus terrificus*	Inhibition of leucocyte migration, dendritic cells maturation, and their IL-6, TNF-α and, IL-12, IL-2, IL-4 and IFN-γ production [12]. Decrease of the expression of MCH II, CD40, CD80, CD86 in dendritic cells [26,28]. Decrease of IgG1 and IgG2a antibodies levels [29].	Immunosuppressive therapy
*Daboia russelii*	Anti- inflammatory response by IL-10 elevation [71].	Immunosuppressive therapy
*Hypanus americanus*	Swelling and leukocyte infiltration in murine models [46].	Study of inflammation
*Lachesis muta muta*	Mast cell activation and degranulation [55].	Study of mast cell dynamics
*Naja kauthia*	Decrease of the excretion of hydroxyproline and glucosamine in the urine, as well as serum levels of acid phosphatase and alkaline phosphatase in arthritic rats [87].	Anti-inflammatory therapy
*Naja mocambica mocambica*	Modulation in cell migration. Increase IL-1B, IL-10, IL-6, TNF- α and eicosanoids release [50]. Activation of the complement system [56].	Study of inflammation dynamics
*Naja naja atra*	Anti-inflammatory response by reducing the levels of TNF-α, IL-1β and the enzymatic activities of myeloperoxidase (MPO) and iNOS synthetase [83,84]. Decrease of the TNF-α, IL-6 and Anti-dsDNA antibodies production [81]. Depletion the number of CD4+ and CD8+ T-lymphocytes [56]. Blocks NF-kB signaling [84].	Immunosuppressive therapy
*Nasonia vitripennis*	Suppression of the inhibitors IkBα and A20 [96].	Study of inflammation dynamics
*Potamotrygon* cf. *henlei*	Neutrophilia dependent on TLR/TRIF signaling [45]	Study of inflammation dynamics
*Tityus discrepans*	TNF, NO production, and morphological alterations in macrophages [38].	Study of inflammation dynamics
*Tityus serrulatus*	Modulation in cell migration [51]. Increase of IL-1B, IL-10, IL-6, TNF- α and eicosanoids release [51]. Stimulate the production of NO and H_2_O_2_ [19]. Suppress the expansion of T-lymphocytes, the synthesis protein and IL-2 production [122].	Study of inflammation dynamics
*Tityus zulianus*	Myeloperoxidase and ROS release [42].	Study of inflammation dynamics
*Trimeresurus flavoviridis*	Activation of the complement system [60].	Study of complement dynamics
*Vipera ammodytes ammodytes*	Stimulates the expression of proinflammatory genes such as Il1a, Il1b, Il-8, Ifna2 and Ifnb1 [73]. Downregulation of the production of IL-12 and IL-18. Inhibition of the NF-kB pathway [74].	Study of inflammation dynamics

TNF-α: Tumor Necrosis Factor-α; IL: Interleukin; MPO: myeloperoxidase; PGE2: prostaglandin E2; and LTB4: Leukotriene B4; IFN: Interferon; TLR: Toll-like receptor; TRIF: Toll/IL-1R domain-containing adaptor-inducing IFN-β.

## Data Availability

Not applicable.

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
