# Peer review of "Biological Effects of Animal Venoms on the Human Immune System"

_toxins, 2022, doi:10.3390/toxins14050344_

Round 1

Reviewer 1 Report

TOXINS Reference manuscript:  Biological effects of animal venoms on the human immune system. Submitted April 2022.

General Comments

In this manuscript the authors are submitting a review on the effects of toxins from venomous animals on the modulation of the human Immune System. Throughout the text, the authors discuss the biological and functional activities of toxins found in different venoms, on cells and on the properties of the human immune system.The objectives of the text, according to the authors, are to better understand the modulation action of different venoms on the immune system, as well as suggestions for possible biotechnological applications. The text is well-written, well-founded, coherent and up-to-date, it is eye-catching. After careful reading, it is my opinion that the text deserves to be published by Toxins, however, I believe that a revised form can be presented, and that it may include some suggestions that may bring more interest to readers and improve the text.

Comments

1- In the Abstract, between lines 4 and 5, and at line 21, the authors wrote … “Venoms are products of specialized glands and serve to many living organisms to immobilize and kill prey, start digestive processes, and sometimes act as a defense mechanism”. Please remove the word sometimes, since an important aspect of animal venoms is act in defense!   2- In the Abstract, between lines 5 and 6, the authors wrote …” Venoms affect different tissues and systems, such as connective tissues and the nervous, coagulation and immune systems”. I suggest rewrite the sentence! …Different cells, cellular structures, and tissues, such as skin, nervous, hematological, digestive, excretory and immune system, as well as heart among other tissue structures.

3- In the Abstract, line 8, the authors wrote  … “Studying the effects on the cells and proteins of the immune system has been”   Better rewrite  ….Studying the effects on the cells and molecules produced by immune system…. Substitute proteins by molecules because the immune system can generate proteins, glycoproteins, lipids and even nucleic acids with biological activities!  

4- In the Abstract, between lines 9 and 10, the authors wrote    ... “The effects of venoms can be the basis for research and development of therapeutic products for different autoimmune diseases”.  I suggest rewrite the sentence by  … can be the basis for research and development of therapeutic protocols useful in modulation of immunological system including different autoimmune diseases. I sugget this because venoms can also be used for instance as adjutants that can increase immune answer and not only autoimmune diseases!    

 5- In The KEYWORDS, the authors wrote    ... “Snake; Scorpion; Bee; envenomation; immunomodulatory; immunosuppression; innate immune system; adaptive immune system. Unless they change the title of the text pointing out only venoms of these animals. Undoubtedly spiders, caterpillars, molluscs, frogs, fishes, among many others venomous animals can modulating biological systems and were not covered in the text.

6- At line 23, the authors wrote …. “Some animals are… Better change to …Venomous animals are…

7- At line 27, the authors wrote… “Poisonous animals coevolved with humans”. In my opinion change to Venomous animals ….

8-  At line 39 … “and in the nineteenth century, with the production of anti-venom sera” … here the authors could include historical references that surely will bring attraction for readers and was the first application of venoms as immunological modulators.  

9- In my opinion the text discussed among lines 33 and 50 could generate a subchapter about “Historical data on venom animals modulating immune system” This could improve the attraction and attention of readers!

10- Between lines134 and 135 the authors wrote      … “Although the interaction between toxins and macrophages has been extensively studied, little is known about the effect of envenomation on neutrophils or polymorpho nuclear cells”. Here the authors undoubtedly need to write about the actions of toxins characterized as phospholipases D from the venoms of spiders of the genus Loxosceles. These molecules act by modulating an indirect activation of leukocytes (neutrophils) and have a well-known mechanism. An indirect endothelium-dependent activation of leukocytes. 

11- Regarding the activation of the inflammatory response, the authors also have a wide field of possibilities discussing insect venoms for instance. 

12- At line 171, the authors wrote    … “which mutates with the production of histamine and lipid mediators, favoring inflammation”  I did not understand the word mutates? From mutation or the authors meant that it resembles? 

13- Among lines 167 and 174. Also the effects of different animal toxins on mast cells are already published and this part of the text could also be more detailed and enlarged!  

14- Effects on complement. Lines 175 to 190. Here again it seemed to me that the authors could have detailed more examples of animal venom toxins acting on the complement system. There are several articles describing actions of toxins on this system that were not described in the text. 

15- In the line 214…. “in models of poisoning with the venom of…” please change poisoning by envenoming

 16- Effects on cytokines  ... This is also a field where several scientific articles studying spider venoms have described activities in the regulation of cytokine production and secretion. Just the authors search at the internet! 

17- Among lines 250 -254 the authors wrote …. “The excretion of hydroxyproline and glucosamine in urine, as well as serum levels of acid phosphatase and alkaline phosphatase, was also decreased in RA models treated with Naja kouthia venom, indicating the protective and anti-inflammatory effect of the on in cartilage degeneration and Toxins 2022, 14, x FOR PEER REVIEW 6 of 17 destruction (87)”.  This part of text is an example of authors to write a subchapter of toxin actions on Extracellular Matrix!  

18-  Effects on B lymphocytes at lines 310 to 324. I believe that this part of the text could be more detailed and elongated. Only two examples of venoms acting on the humoral immune response seem little to me! I am sure that if authors perform a more detailed search at internet they will find more examples of toxins modulating such as activities. 

19- In the line 327   … “adhesion molecules, such as LFA-1 and VLA-4, I suggest the authors use the most accepted terms in the specialized literature right after LFA1 (integrin alpha L Beta 2) and VLA-4 (integrin Alpha 4 Beta-1)

 20- The authors could research on venoms and/or purified toxins acting on bone marrow cells. This could enrich the review! 

21-  The authors wrote between lines 366-368 … “Other toxins have been shown to stimulate the differentiation of lymphocyte subpopulations, such as those derived from the conotoxins of the marine snail Californiconus cali fornicus called cal14.1b and cal14.2c (120)”.  One more reason for the authors to complete their Keyword or sentences throughout of text with more examples of animals besides the ones already mentioned! 

22- At lines 378 a 380… “This effect on the inhibition of the Th17 response could serve as the basis for the development of therapeutic agents against pathologies associated with this subpopulation, such as multiple sclerosis, inflammatory bowel disease, psoriasis and RA (120,121)”. Phrases like this could be part of a subchapter where possible biotechnological applications were postulated. 

23- The text presented in the conclusions is repetitive and has already been discussed in the introduction and in other parts of the text. Do not include anything new! In my opinion, the authors could substitute this text for possible biotechnological applications of the studied venoms and toxins.    

24- Among lines 264 to 270…. “Effects on inflammasomes”  here the authors undoubtedly need to include a discussion of phospholipase D from the venoms of Loxosceles spiders, which have a very marked pro-inflammatory activity. If not the most potent ones already described in the literature! and that surely have activities in the innate immune response. 

25- I would also include a subchapter on the actions of toxins on endothelial cells with anti-angiogenic or angiogenic potential. 

26- I would also include a subchapter on the actions of toxins on Extracellular Matrix structure, organization, and components! Fibrous glycoproteins, basement membrane molecules, proteoglycans, glycosaminoglycans, Hyaluronic acid, are some examples of possibilities! Venom animal proteases, Lyases and Hyaluronidases are some examples that could modulate the immune system! and can be included in a revised text. 

27- Disintegrins (toxins found in the venoms of several snakes), which are molecules that act by modulating the inhibition of platelet adhesion and agregation, by inhibiting  integrin binding to the extracellular matrix, seem to be excellent examples of toxins modulating cell migrations and the immune system. 

28- I also felt a lack of indications of venoms or toxins that could act with an adjuvant effect. For example by stimulating leukocyte attraction at the application site and thus facilitating the immune response. 

29- I missed the authors including clear examples of biotechnological possibilities of uses of animal toxin in the text. As examples of useful tools in the treatment or diagnosis of autoimmune diseases, or immune system modulators! 

30- Finally, the authors could include in the text some figures pointing representative discussed venomous animals, some clinical effects after envenomation, and a figure with putative suggestions for biotechnological applications based on venoms/toxins properties, which surely will attracted readers attention!       

Reviewer 2 Report

The manuscript entitled “Biological effects of animal venoms on the human immune system” gives an interesting overview on biological effects of diverse venom on the human immune system and how some of venom components can be useful for the study and development of immunomodulatory drugs. Although there are several topical reviews covering different aspects regarding therapeutic potential of venom and its components as well as their effects on different systems, this specific filed is still attractive and there are more and more papers published each year. In line with that there is always space for another updated overview.

In paragraph 3.2. Effects on T lymphocytes, when talking about the effects on peripheral blood lymphocytes, authors could mention a few more studies and the underlying mechanisms of venoms (e.g. bee venom) as well as its active peptide components (such as melittin) on lymphocyte cyto/genotoxicity. Both bee venom and melittin are cytotoxic for lymphocytes and are able to induce morphological changes in the cell membrane, granulation and lysis of the cells. Besides, they show increased DNA damage including oxidative DNA damage as well as increased formation of other markers of genomic instability. This genotoxicity coincides with increased formation of ROS, reduction of GSH, increased LPO as well as PLC activity, showing the induction of oxidative stress. Melittin itself is also capable of modulating gene expression patterns of genes involved in DNA damage response, oxidative stress and apoptosis.

Garaj-Vrhovac V, Gajski G. Evaluation of the cytogenetic status of human lymphocytes after exposure to a high concentration of bee venom in vitro. Arh Hig Rada Toksikol. 2009; 60(1): 27-34.

Gajski G, Garaj-Vrhovac V. Bee venom induced cytogenetic damage and decreased cell viability in human white blood cells after treatment in vitro: a multi-biomarker approach. Environ Toxicol Pharmacol. 2011; 32(2):201-11.

Gajski G, Domijan AM, Garaj-Vrhovac V. Alterations of GSH and MDA levels and their association with bee venom-induced DNA damage in human peripheral blood leukocytes. Environ Mol Mutagen. 2012; 53(6): 469-77.

Gajski G, Domijan AM, Žegura B, Štern A, Gerić M, Novak Jovanović I, Vrhovac I, Madunić J, Breljak D, Filipič M, Garaj-Vrhovac V. Melittin induced cytogenetic damage, oxidative stress and changes in gene expression in human peripheral blood lymphocytes. Toxicon. 2016; 110:56-67.

Minor remarks:

Page 1, line 20 – change to “Venoms are produced by animals, plants or microorganisms as an evolutionary…”

Page 4, line 164 – do authors mean “Naja mozambique mozambique” as in Mozambique spitting cobra?

Page 4, line 185 – full stop is needed after B – B. jararacussu

Page 5, line 252 please revise “…indicating the protective and anti-inflammatory effect of the on in cartilage degeneration and destruction (87).”
